# Developing a Support Vector Regression (SVR) Model for Prediction of Main and Lateral Bending Angles in Laser Tube Bending Process

**DOI:** 10.3390/ma16083251

**Published:** 2023-04-20

**Authors:** Mehdi Safari, Amir Hossein Rabiee, Jalal Joudaki

**Affiliations:** Department of Mechanical Engineering, Arak University of Technology, Arak 38181-41167, Iran

**Keywords:** laser tube bending process (LTBP), support vector regression (SVR), bending angles

## Abstract

The laser tube bending process (LTBP) is a new and powerful manufacturing method for bending tubes more accurately and economically by eliminating the bending die. The irradiated laser beam creates a local plastic deformation area, and the bending of the tube occurs depending on the magnitude of the heat absorbed by the tube and its material characteristics. The main bending angle and lateral bending angle are the output variables of the LTBP. In this study, the output variables are predicted by support vector regression (SVR) modeling, which is an effective methodology in machine learning. The SVR input data is provided by performing 92 experimental tests determined by the design of the experimental techniques. The measurement results are divided into two sub-datasets: 70% for the training dataset, and 30% for the testing dataset. The inputs of the SVR model are process parameters, which can be listed as the laser power, laser beam diameter, scanning speed, irradiation length, irradiation scheme, and the number of irradiations. Two SVR models are developed for the prediction of the output variables separately. The SVR predictor achieved a mean absolute error of 0.021/0.003, a mean absolute percentage error of 1.485/1.849, a root mean square error of 0.039/0.005, and a determination factor of 93.5/90.8% for the main/lateral bending angle. Accordingly, the SVR models prove the possibility of applying SVR to the prediction of the main bending angle and lateral bending angle in LTBP with quite an acceptable accuracy.

## 1. Introduction

Tube bending is important in manufacturing. When the tube thickness is thin, the applied force or moment can deform the cross-section from its initial circular shape into an elliptical cross-section, and ovalization appears as an unwanted phenomenon [1]. The use of a laser beam in this process has several benefits: by using a laser beam, no mechanical load will be applied to the parts, and the irradiated heat/energy can be controlled very easily [2]. The cost of die fabrication will be waived, and a precise bending angle can be obtained by springback compensation and the re-arranging of the process parameters. Springback is the main phenomenon in tube bending. This phenomenon happens due to elastic recovery in the unloading stage of the tube. The stress and strain vary from a positive magnitude from the outside of the bend area to a negative magnitude in the inside area of the bend zone. After unloading, the bend angle and bend radius vary by releasing the elastic strain created in the loading stage. One of the main challenges in bending, especially in tube bending, is the prediction of and compensation for springback. The springback compensation can be carried out by predicting the bending angle with an analytical solution or statistical prediction [3].

The first attempt to predict the bending angle occurred in the year 2000, by Cheng and Lin [4] and Dragos et al. [5]. Casalino and Ludovico [6] were the first researchers that used a trained neural network to predict the process variables of laser sheet bending to reduce springback. Barletta et al. [7] and Gisario et al. [8] utilized an artificial neural network (ANN) tool for the decrease and compensation of springback in a laser-assisted bending process. Initially, an analytical model was developed to foretell the springback according to the parameters of the forming process. Then, the trained ANN helped to increase the correlation quality of the developed network. Thus, by combining ANN and an analytical solution, a powerful tool was developed to control the springback more efficiently. Lambiase et al. [9] investigated the bending of thin 304 stainless steel sheets by laser beam irradiation using ANN. The effects of the process variables, including the laser power, the number of passes, the scanning speed, the cooling time on the bending angle and the maximum temperature, were investigated by constructing and training the ANN. The trained network can control and automate the laser bending process by applying optimal conditions of the bending process. Imhan et al. [10] studied the laser bending of AISI 304 stainless steel tubes. The bending angle was calculated by an analytical model. The analytical model considered the variation in material properties from the increase in temperature. Then, particle swarm optimization (PSO) was employed to optimize the results and to decrease the mean absolute error. The PSO optimization helped to determine values for the material specification (density, yield stress, elastic modulus, specific heat, and coefficient of thermal expansion (CTE)) such that the effect of the temperature increase was included in the analytical model. Fetene et al. [11] studied the effects of the width and thickness of the sheet in laser bending AH36 steel. The effect of process variables (scanning speed, laser power, laser beam diameter, thickness, width of strip, and the number of passes) was investigated using Taguchi’s L27 array. An analysis of variance (ANOVA) was utilized to assess the effectiveness of the process variables, and a regression equation was proposed for the bending angle prediction. The optimized parameters were obtained by the Taguchi method. Thus, by analyzing the signal-to-noise ratio of the measured results, the optimum bending angle can be determined. Using metaheuristics algorithms for the prediction of the process parameters of laser bending is mostly common in laser sheet bending, and few research studies have been conducted using a prediction of laser tube bending. Fetene et al. [12] used the combination of the finite element method (FEM) and ANN for the prediction of the laser-assisted bending of an AA5052-H32 aluminum alloy sheet. After verifying the FEM model by experimental tests, several laser bending conditions were modeled (by varying the power, external load, scan speed, and distance of the irradiation line from the free end), and the data were extracted. Then, the ANN model was used to predict the bend angle. The results showed that the ANN can be used for the prediction of the bending angle. Moreover, similar research was conducted and published by Kant et al. [13] for the estimation of absorptivity in the laser bending process.

The genetic algorithm (GA) is another popular tool for prediction in the soft computing field. However, due to the huge data required for prediction by the GA, it cannot be used by itself, and the GA is usually used with the combination of ANN or neuro-fuzzy systems. Maji et al. [14,15,16] compared the effectiveness and performance of the GA with ANN (GA-NN) and an adaptive neuro-fuzzy inference system (GA-ANFIS) in laser bending. A batch of experimental data was used to assess both approaches. Both approaches predicted the bending angles effectively. The results show that the GA-NN approach could predict the bending angles more precisely than the GA-ANFIS approach. Also, GA-NN (using the genetic algorithm as training function) was compared with ANN (using back-propagation algorithm as training function), and it was proved that the GA performs better than the back-propagation algorithm (with a higher accuracy) for the prediction of the bending angle by both the networks. In a similar study, Keshtiara et al. [17] used ANN and the GA to predict the bending angle of the laser tube bending process and the optimization of the process parameters. The utilization of ANN, GA, ANFIS, PSO, and other predicting tools is associated with developing MATLAB 2017 software.

Safdar et al. [18] studied the bending of AISI 304L stainless steel tubes with different laser scanning strategies. The three scanning strategies were axial scanning, alternate circumferential scanning, and the sequential circumferential scanning scheme. The results demonstrated that the axial scanning scheme creates twice the bending angle and less distortion of the inner edge compared to other scanning strategies. Furthermore, an alternate circumferential scanning scheme led to decreased lateral bending compared to lateral bending, which normally occurs for a circumferential scanning scheme. Wang et al. [19] developed an algorithm for bending tubes in 2D and 3D space. The algorithm consists of dividing the tube into several segments based on the extremum and the inflection point of the expected tube shape. The scanning path is then determined according to the derivative and second derivative of the desired shape of the tube. Additionally, for the 3D bending of the tube, the 3D curve was decomposed into two 2D curves, and the desired shape was achieved by combining the 2D path data. As can be seen, the formation of the tube in 2D and 3D space is very complicated, and closed-form algorithms can help the manufacturer with the fabrication of complicated shapes. Nath and Yadav [20] used the integral transform technique (an inverse solution) to solve the 3D heat transfer equation during laser beam irradiation to find the final bending angle by predicting the strain field distribution while heating and cooling. By the inverse estimation of the absorption coefficient, the analytical model was developed, and the temperature field was determined. In addition to tube bending, the irradiated heat from the laser source can create local strengthening in the tube. Kim and Yoon [21] showed that the strengthening happened in the cross-sectional plane perpendicular to the axial direction of the tube. In addition, the strengthening was more effective when the number of folds was increased and the energy absorption was maximized by up to 32.2% during axial crushing by a helical laser beam irradiation pattern.

Support vector regression (SVR) is a learning algorithm implemented for the estimation of discrete values according to the principle of support vector machines (SVM). SVM was introduced by Vladimir Vapnik [22] in 1992, and is a popular machine learning tool for classification and regression. SVR is a powerful tool for estimation. Several research works were published in different areas of mechanical engineering that utilized SVR. SVR can predict the drilling force drilling an internal hole in carbon-fiber-reinforced polymer (CFRP) [23], provide a wear prediction for cutting tools in the milling process [24], predict cutting force and temperature in bone drilling [25], predict cutting force and surface roughness in the milling process [26] and turning process [27], predict the laser cutting process cost for AISI316L stainless steel [28], and so on. By reviewing the different utilizations of the SVR model and comparing the prediction results from other metaheuristic approaches (such as ANN, GA, and ANFIS, etc.) it can be concluded that the SVR model can predict the output parameters in complicated systems more accurately.

To the best of the literature review and our knowledge, there is no published research work on modeling and predicting the laser tube bending process (LTBP) using SVR. This research aims to use the support vector regression method as one of the good approximators in machine learning techniques to predict the main and lateral bending angles. The basic idea behind SVR is to find the line of best fit. In conventional curve-fitting approaches, a straight line is fitted by the least square method. Each point has an equal portion in determining the slope of straight line by the least square method, while the attitude of new curve-fitting procedures is different. In new approaches, a weight function will be associated with each data point, and higher priority will be considered for points closer to the fitted line. The authors utilized the SVR model due to its motivating features, such as scattered solution representation, lack of local minimums, good generalization performance, and strong theoretical basis. The model predictor in this article will calculate the process outputs (the main bending angle and lateral bending angle) based on the input process variables, including the laser power, laser diameter, scanning speed, irradiation length, irradiation scheme, and the number of irradiations. The effect of process variables was analyzed and discussed by the authors in a previous study [29], and this article will focus only on the implementation of SVR for the prediction of process variables. It is recommended that the readers read the authors’ previous publication [29] for additional details and then read this article.

## 2. Modeling and Data

### 2.1. Laser Tube Bending Process and Data Preparation

Figure 1 shows a schematic view of the LTBP, with the definition of the main bending angle along the vertical direction and the lateral bending angle in the horizontal plane. To find a regression equation, it is necessary to prepare the initial data, which determine the output parameters (the main bending angle and lateral bending angle) by the variation of the input variables (independent variables) of the process. The basics of SVR will be explained in the next section. The laser tube bending was carried out by a set of experiments to prepare the required data for SVR. The experiments were carried out on a tube made of mild steel (chemical composition: Fe base, 0.23% C, 0.20% Cu, 0.83% Mn, 0.04% P, 0.05% S, and 0.28% Si) and with an 18 mm outer diameter and a thickness of 1 mm. The tube was cut into samples 100 mm in length. The tube samples were clamped from one side, and the laser irradiated along the longitudinal direction (axial irradiating scheme (AIS)) or tangential direction (circular irradiating scheme (CIS). The LTBP process variables were the laser power, laser beam diameter, scanning speed, irradiation length, irradiation strategy, and the number of irradiations. The ranges of process variables are shown in Table 1. The experimental tests were carried out according to the response surface methodology plan (Box–Behnken type). The plan consists of 92 experiments, which are listed in Table A1, Appendix A. Minitab 19 software was used for the design of experiments and data analysis in the published work [29]. After the irradiation of the laser beam, the tube bent upward (main bending angle). Unfortunately, due to non-homogenous heating along the irradiation path, a small in-plane deviation was observed, which is called the lateral bending angle. The length of irradiation was equal in the CIS and AIS at each level. To attain the SVR model, the input data was normalized in the range of [0, 1]. More details of the experimental setup and definitions are presented in Ref. [29] and are not repeated in this article. 

### 2.2. The Basics of Support Vector Regression (SVR)

SVR is a powerful tool for the solution of scattered problems in which local minimums cannot be found by other regression methods. The SVR method is a type of nonlinear classification method which categorizes the results of tests among several groups. By defining a nonlinear classification threshold from data of N independent variables mapped to a space that has more dimensions and utilizing training data ${\left\{\left(z_{k},q_{k}\right)\right\}}_{k=1}^{N}$, a linear model can be developed [22] where z is the independent variable and q is the classification vector, defined $q_{k}\in \left\{-1,1\right\}$. The classification is performed for N samples. The classification is performed by defining an optimum hyperplane defined by Equation (1). The optimum hyperplane must meet the conditions of Equations (2) and (3).
(1)$$g^{T}\emptyset \left(z_{k}\right)+h=0$$
(2)$$g^{T}\emptyset \left(z_{k}\right)+h\geq 1\;para\;q_{k}=1$$
(3)$$g^{T}\emptyset \left(z_{k}\right)+h\geq -1\;para\;q_{k}=-1$$
where g is the weight vector. Equations (1)–(3) define the basics of SVR. However, for practically implementing the SVR method to predict the output parameters by regression between the input parameters, a mapping function should be defined. The mapping function $\emptyset \left(\cdot \right):R^{n}\to R^{n_{k}},$ from independent variables to a space with larger dimensions, is defined by combining Equations (1)–(3) and is written as Equation (4):(4)$$q_{k}[g^{T}\emptyset \left(z_{k}\right)+h]\geq 1$$

Equation (4) was presented by Vapnik [22] to classify the samples into two classes which are not applicable to obtaining the optimal hyperplane equation. In this way, a tolerance margin β≥0 at the classification threshold is added to Equation (4), which is written as Equation (5). The minimum of the g function for finding the optimal hyperplane can be calculated by Equation (6).
(5)$$q_{k}\left[g^{T}\emptyset \left(z_{k}\right)+h\right]\geq 1-\beta_{k}$$
(6)$$\begin{array}{c} min\\ g,h \end{array}\frac{1}{2}{\left\|g\right\|}^{2}+C\sum_{k=1}^{N}\beta_{k}$$
where C is the tuning parameter of optimum hyperplane classification of Equation (5). Patel et al. [30] and Huang and Tsai [31] showed that using $f\left(z,g\right)=g^{T}z+h$ a linear regression function can result in better predictions. In this way, a threshold error, which is called the ϵ-insensitivity loss function, is utilized. If the ϵ-insensitivity loss function is defined as Equation (7), the difference between the classification vector *q* and the optimum hyperplane settles in the bandwidth between the $-\epsilon -\beta_{i}^{*}$ and $\epsilon +\beta_{i}$ (Equation (8)).
(7)$${\left|q-f\left(z,g\right)\right|}_{\epsilon }=\left\{\begin{array}{c} 0,\\ \left|q-f\left(z,g\right)\right|-\epsilon , \end{array}\right.\begin{array}{c} case\left|q-f\left(z,g\right)\right|\leq \epsilon \\ otherwise \end{array}$$
(8)$$-\epsilon -\beta_{i}^{*}{\leq (g}^{T}z+h)-q_{i}\leq \epsilon +\beta_{i}$$

β and $\beta^{*}$ are equal in symmetric conditions. The residual function of minimization is defined as Equation (9), and SVR searches to find the minimum residual in the bandwidth of β and $\beta^{*}$.
(9)$$R=\frac{1}{2}{\left\|g\right\|}^{2}+C\sum_{i=1}^{N}\left(\beta_{i}+\beta_{i}^{*}\right)$$
where *R* is the cost function. The tuning parameters should be defined such that they minimize the system output. To determine these parameters, the data set of results is divided into two subsets. The first sub-set is used for training and finding the optimal value, and the second sub-set is used for validating the obtained parameters by finding the smallest possible error. The equations of the SVR model were derived by Vapnik [22], Patel et al. [30], and Huang and Tsai [31].

The prediction error can be calculated in different ways. Four common definitions of error equation are the mean absolute error (MAE), mean absolute percentage error (MAPE), root mean square error (RMS), and determination factor (*R*^2^), which are demonstrated in Equations (10)–(13). $\theta_{i}$ and ${\hat{\theta }}_{i}$ are the measurement data and predicted magnitude of output by SVR, respectively. Additionally, $\theta_{i}^{mean}$ and ${\hat{\theta }}_{i}^{mean}$ are the mean of each parameter, respectively. These four error definitions show different aspects of error. The MAE and RMS are absolute values of error, while the MAPE is a relative error definition. The determination factor differs from the three other equations. The determination factor shows the quality of correlation between the real measured data and the value predicted by the regression equation. A more precise correlation will be obtained for the value of the determination factor nearer to 100%. To evaluate the goodness of the SVR model, it is essential to include all error definitions in consideration.
(10)$$MAE=\frac{1}{n}\sum_{i=1}^{n}\left|\theta_{i}-{\hat{\theta }}_{i}\right|$$
(11)$$MAPE=\frac{1}{n}\sum_{i=1}^{n}\left|\frac{\theta_{i}-{\hat{\theta }}_{i}}{\theta_{i}}\right|\times 100\%$$
(12)$$RMSE=\sqrt{\frac{1}{n}\sum_{i=1}^{n}{\left(\theta_{i}-{\hat{\theta }}_{i}\right)}^{2}}$$
(13)$$R^{2}=\frac{{\left[\sum_{i=1}^{n}\left(\theta_{i}-\theta_{i}^{mean}\right)\left({\hat{\theta }}_{i}-{\hat{\theta }}_{i}^{mean}\right)\right]}^{2}}{\left[\sum_{i=1}^{n}\left(\theta_{i}-\theta_{i}^{mean}\right)\right]\left[\sum_{i=1}^{n}\left({\hat{\theta }}_{i}-{\hat{\theta }}_{i}^{mean}\right)\right]}$$

Figure 2 illustrates the flowchart of the attained SVR predictor model. The SVR models erre developed with Python 3.8 software. As mentioned in Section 2.1, 92 experimental tests were carried out based on the variation of six important LTBP process variables. The process variables are listed in Table 1, and the two output variables are the main bending angle and lateral bending angle. After implementing the experimental tests and measuring the output variable, the results should be split into two sub-sets for cross-validation. In this article, the holdout cross-validation technique (70% training data; 30% testing data) was used to evaluate the accuracy of the SVR model. Thus, the obtained results were randomly divided into two sub-sets: 70% of the results were used for training, and the remaining 30% were used to assess the accuracy of the SVR model. The SVR model was developed and trained by the determined sub-sets of data. The prediction was carried out in Python software, and the results were plotted using MATLAB software.

## 3. Results and Discussion

As explained earlier, the data set was obtained during the process of experimental tests. After preprocessing, the dataset was divided into two sub-datasets of training (70%) and testing (30%) data. The SVR model was attained during the training process, and it was then evaluated by the test sub-dataset. It is worth mentioning that the hyperparameters (C and ϵ) were changed in the training phase, and in the final model, values (12, 0.1) and (9, 0.1), were used to estimate the main and lateral bending angles, respectively.

In this study, the independent variables were the input variables of the model, including the laser power, laser beam diameter, scanning speed, irradiation length, irradiation scheme, and the number of irradiation passes. Additionally, the dependent variables were the output variables (the main bending angle and lateral bending angle), which were estimated based on the inputs. The main goal of the presented work was to obtain a predictor model that can accurately estimate the desired outputs based on the input values.

Figure 3 shows the comparison of the results predicted by the SVR model and the measured data for the main bending angle and lateral bending angle separately. The blue circles and red squares signify the train data and test data, respectively. A line with a 45° slope, which shows the A=P line (*A*: actual data; *P*: predicted by SVR model), is illustrated in Figure 3. A better correlation between the actual and predicted data will be obtained for closer points to the line. The determination factor R^2^ is 0.98 and 0.96 for the total data (the training and test sub-sets) of the results of the main bending angle and lateral bending angle, respectively. Despite the fact that the determination factor is higher for the main bending angle results, the quality of the correlation for both output parameters is quite acceptable. It should be noted that the output parameter is the function of six independent process variables and their interactions. Therefore, finding a regression line is very difficult with common regression methods. The difference between the predicted value and the actual value is very low, demonstrating the quality of the correlation.

Figure 4 and Figure 5 illustrate the absolute error and the MAPE error for the main bending angle and lateral bending angle, respectively, for all experimental tests. As can be seen, the blue and red bars represent the training data (70% of total results) and the test data. For the main bending angle, the absolute error at the training stage is very close to zero. The maximum absolute error is 0.027. Similarly, the absolute error is low for the test data, and the maximum absolute error is 0.15. The MAPE shows the same trend as the absolute error such that the maximum MAPE for the training and test data is 1.87% and 10.38%, respectively.

Although the absolute error for the lateral bending angle is lower than similar values for the main bending angle (which is due to the lower lateral angle magnitudes than the main angle), a higher MAPE was observed. The maximum MAPE is 3.18% and 7.04% for the training data and the test data, respectively. However, more error is observed in assessing the testing data of the lateral bending angle than the main bending angle.

For a better understanding of the accuracy of the SVR model, the real and predicted values of the main bending angle and lateral bending angle were plotted in Figure 6. The real training data and the test data are illustrated by the solid blue line and solid red line, respectively. The predicted training data and predicted test data are demonstrated with blue circles and red squares, respectively. The blue circles are placed on the blue solid line for both outputs, which specifies the higher accuracy of the SVR model in the training stage. The SVR model was created based on the training dataset; thus, the readers should look at the trials in the test stage for a better understanding of the accuracy of the model, which are the data that the SVR had not encountered before. Comparing the graphs in Figure 6 demonstrates that the accuracy of the SVR model is also high for the prediction of the lateral bending angle. In addition, the model performance is relatively better for the prediction of the main bending angle.

Table 2 shows the MAE, MAPE, and RMSE and the determination factor, R^2^, for the training and test stages of the main bending angle and lateral bending angle, respectively. The results show that both the MAE and RMSE are close to zero and very small for both the main bending angle and lateral bending angle predictions. Similarly, the MAPE is very low for both outputs. Furthermore, the determination factor, R^2^, is above 90%, which shows the goodness of the accuracy of the SVR model in the prediction of the laser bending process output variables. Different values of error illustrate that the accuracy of the SVR model is relatively higher in the prediction of the main bending angle than the lateral bending angle. The bending angle is about 10 times higher than the lateral bending angle. The lateral bending angle is less than 0.20°, which requires a very accurate approach to be estimated. The MAE and RMSE values of Table 2 show that the error of prediction is as low as 1.5% (relative error).

The difference between the actual and the predicted data for the main bending angle and lateral bending angle is illustrated as a histogram in Figure 7. As can be seen, the accumulation of error is high around the zero-point prediction, which can be called a singularity point in the bending angle prediction. For the main bending angle, a rare prediction was founded outside the range of −0.05≤error≤0.05. Likewise, the range for the lateral bending angle is −0.005≤error≤0.005. Lastly, it is worth noting that the current work is the first attempt to implement the SVR method for the prediction of the main bending angle and lateral bending angle as a very complicated process variable that depends on six different input variables. The accuracy and precision of a metaheuristic methods depend on the volume of investigation data. In this work, 92 sets of experiments were carried out, and the measured data were included in the analysis. Increasing the amount of data may lead to a better prediction, though the current model has an acceptable accuracy. 

## 4. Conclusions

In this study, a modified support vector machine algorithm called support vector regression was used to predict the main bending angle and lateral bending angle in the laser tube bending process. The SVR model was trained using 70% of the experimentally measured data and by using hyperparameter (C, ϵ) values of (12, 0.1) and (9, 0.1) as the main and lateral bending angles estimated, respectively. The results show that the SVR model can predict the process output of laser tube bending very well. The MAE, RMSE, and MAPE were calculated for the main bending angle and lateral bending angle as 0.021, 0.039, and 1.485%, demonstrating the effectiveness of the trained SVR model. Likewise, the mentioned error criteria for the lateral bending angle were calculated as 0.003, 0.005, and 1.849%. The determination factor was 93.5% and 90.8% for the prediction of the main bending angle and lateral bending angle, respectively. The lateral bending angle was less than 0.20°, which requires a very accurate approach to be estimated. The MAE and RMSE values show that the error of prediction is as low as 1.5%. As a final point, the SVR model performed better in predicting the main bending angle when compared to the lateral bending angle due to the lower MAPE value and higher determination factor.

## Figures and Tables

**Figure 1 materials-16-03251-f001:**
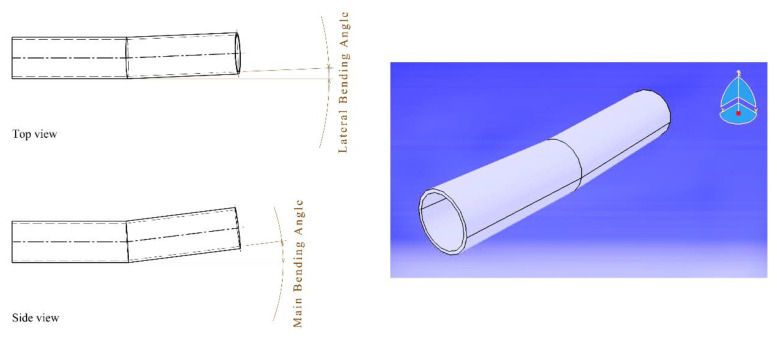
Schematic view of LTBP and definition of main and lateral bending angles.

**Figure 2 materials-16-03251-f002:**
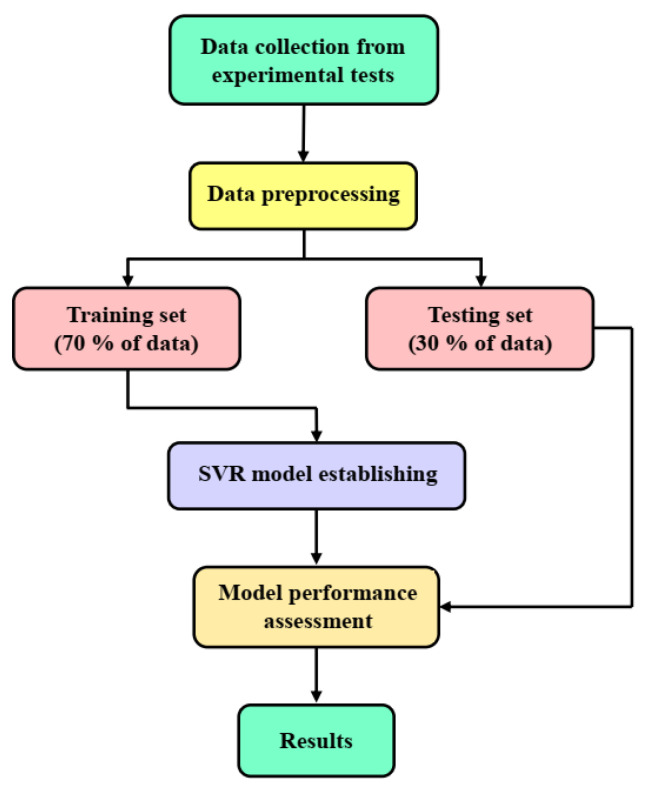
Flowchart of attained SVR predictor model.

**Figure 3 materials-16-03251-f003:**
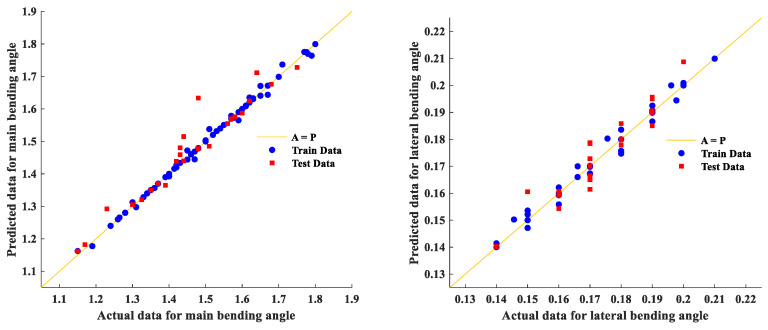
The actual measurements vs. the results predicted by the SVR model for the main and lateral bending angles (all angles in degrees).

**Figure 4 materials-16-03251-f004:**
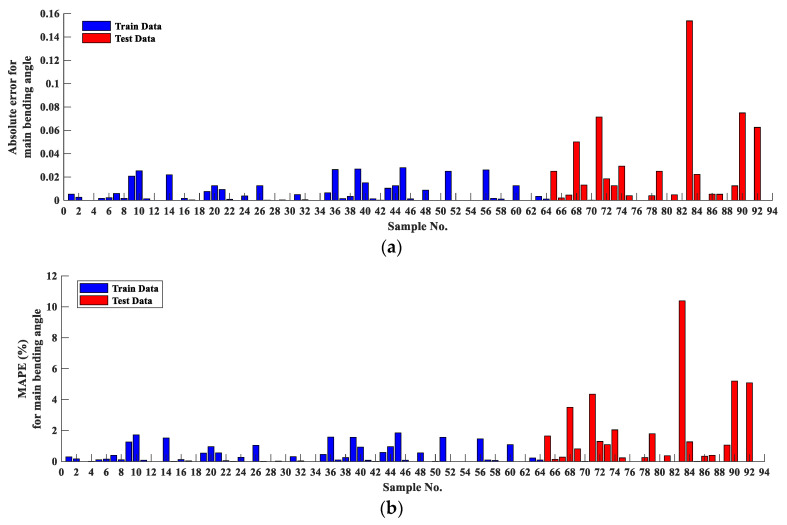
(**a**) Absolute error; (**b**) mean absolute percentage error (MAPE) for all results corresponding with the main bending angle.

**Figure 5 materials-16-03251-f005:**
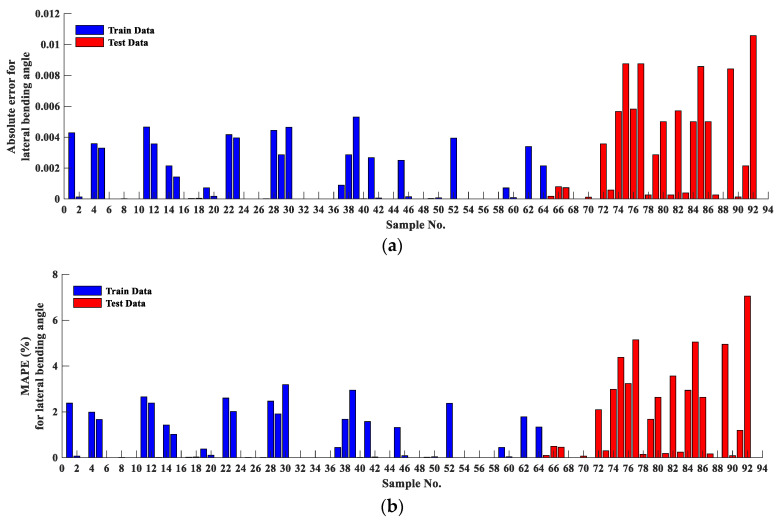
(**a**) Absolute error; (**b**) mean absolute percentage error (MAPE) for all results corresponding with lateral bending angle.

**Figure 6 materials-16-03251-f006:**
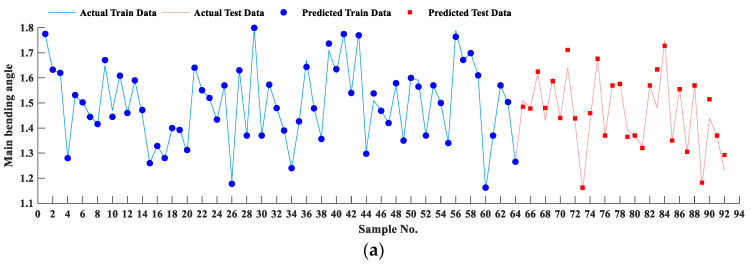
The actual and predicted values for (**a**) main bending angle and (**b**) lateral bending angles for all samples in both training and test phases.

**Figure 7 materials-16-03251-f007:**
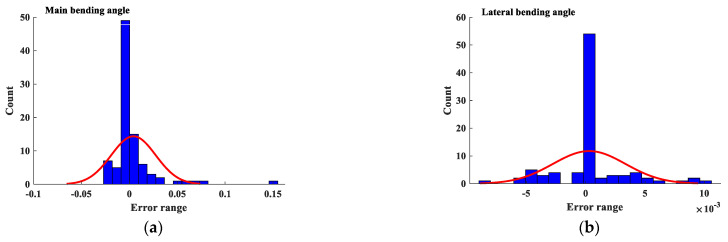
Histogram of the error between the actual and the predicted data for the (**a**) main bending angle and (**b**) the lateral bending angle.

**Table 1 materials-16-03251-t001:** The levels of process parameters variation in experimentation [29].

Process Parameters	Value	Unit
Laser Power	500 800 1100	W
Scanning Speed	10 15 20	mm/min
Laser Beam Diameter	4 6 8	mm
Irradiation Length in CIS in AIS	20 100 180 3.14 15.70 28.27	degree mm
Number of Irradiations	1 3 5	-
Irradiation Scheme	Circular irradiating scheme (CIS) Axial irradiating scheme (AIS)	-

**Table 2 materials-16-03251-t002:** The calculated MAE, MAPE, RMSE, and determination factor (R^2^) in the training and test phases for main and lateral bending angles.

Process Output	Data Set	MAE	MAPE (%)	RMSE	R^2^
Main bending angle	Train	0.005	0.365	0.010	0.990
	Test	0.021	1.485	0.039	0.935
Lateral bending angle	Train	0.001	0.632	0.002	0.981
	Test	0.003	1.849	0.005	0.908

## Data Availability

Data is contained within the article.

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
