# Peer review of "Developing a Support Vector Regression (SVR) Model for Prediction of Main and Lateral Bending Angles in Laser Tube Bending Process"

_materials, 2023, doi:10.3390/ma16083251_

Round 1

Reviewer 1 Report

Check that the abstract provides an accurate synopsis of the paper. It is very vague in present form.

Methodology of the proposed model must be illustrated by a clear flowchart.

Besides, the writing of the paper, including contributions, methodologies, should be clearer and highlight the innovation of methods & principles. I can see that the best fit curve is just a straight line owing to linear association between dependent and independent parameter and can be fitted using simple linear regression using least square technique. So what is the necessity of using SVR and machine learning? Novelty is not clear.

On the similar lines stated above, insufficient literature is presented to support the aim of the study. This point still needs further revision.

Results are presented in wrong way. Regression is used to find association between dependent and independent parameter. Figure 1 presents the actual measurements vs. predicted results by the SVR model for the main and lateral bending
angle
s. Which one is dependent and independent parameter here? Actual and predicted values of same angle are plotted on X and Y axis. This is just a comparison and not regression.

This paper doesn’t illustrate procedure of data collection, preparation, pre-processing, and processing in detail.

Was the data normalized/ standardized?

Additionally results must be provided considering different holdout % and holdout validation approach.

There is no discussion on results. The results are also included considering limited examples.

Was the algorithm trained using standard hyperparameters or were they altered?

Comment on computational time and complexity in training of algorithm.

The manuscript is more like a report rather than a research paper failing in solid discussion. Revise results and discussion part by critically examining results and include inferences drawn. 

How to ensure robustness of the model in high noisy environment?

How to deal with the data diversity of present moment and moment in the future?

Author Response

Reviewer #1:

 The authors greatly appreciate the reviewer for their insightful comments and suggestions. According to the reviewer comments, the manuscript has been edited. The comments were sorted, and the relevant answers were highlighted in green color as follows. The necessary changes in the manuscript have been shown in red color.

- Check that the abstract provides an accurate synopsis of the paper. It is very vague in present form.

The abstract was edited as follows and some sentences are added to the manuscript.

Abstract: The laser tube bending process (LTBP) is a new and powerful manufacturing method for bending tubes more accurately and economically by eliminating the bending die. The irradiated laser beam creates a local plastic deformation area and the bending of the tube occurs depending on the magnitude of absorbed heat by the tube and material characteristics. The main bending angle and lateral bending angle are the output variables of the LTBP. In this study, the output variables will be predicted by support vector regression (SVR) modeling which is an effective methodology in machine-learning. The input data of SVR is provided by performing 92 experimental tests determined by the design of the experiments techniques. The measurement results are divided into two sub-datasets: 70% training dataset, and 30% testing dataset. The inputs of the SVR model are process parameters, which can be listed as laser power, laser beam diameter, scanning speed, irradiation length, irradiation scheme, and the number of irradiations. Two SVR models are developed for the prediction of the output variables separately. The SVR predictor achieved a mean absolute error of 0.021/0.003, mean absolute percentage error of 1.485/1.849, root mean square error of 0.039/0.005 and determination factor of 93.5/90.8% for main/lateral bending angle. Accordingly, the SVR models proved the possibility of applying SVR in the prediction of the main bending angle and lateral bending angle in LTBP with a quite acceptable accuracy.

- Methodology of the proposed model must be illustrated by a clear flowchart.

Appreciating the comments of the respected reviewer, a clear flowchart of the presented SVR model was added in the revised manuscript as follows:

Figure 1. Flowchart of attained SVR predictor model.

In addition, explanations related to the flowchart were also added in the revised manuscript as follows:

Fig. 1 illustrates the flowchart of attained SVR predictor model. As explained earlier, the data set is obtained during the process of experimental tests. After preprocessing, the dataset is divided into two sets of 70% training and 30% testing sub-datasets. The SVR model is attained during the training process and then it is evaluated by the test sub-dataset.

- Besides, the writing of the paper, including contributions, methodologies, should be clearer and highlight the innovation of methods & principles. I can see that the best fit curve is just a straight line owing to linear association between dependent and independent parameter and can be fitted using simple linear regression using least square technique. So what is the necessity of using SVR and machine learning? Novelty is not clear.

Appreciating the scrutiny of the respected reviewer, we used the SVR model due to its motivating features such as scattered solution representation, lack of local minimums, good generalization performance, and strong theoretical basis. In conventional curve fitting approaches a straight line is fitted by the least square method. Each point has an equal portion in determining the slope of straight line by the least square method while the attitude of new curve fitting procedures is different. In new approaches, a weight function will be associated with each data and higher priority will be considered for closer points to the fitted line. To the best of the literature review and our knowledge, there is no paper about modeling and predicting of laser tube bending process (LTBP) using support vector regression (SVR), as one of the good approximators in machine learning techniques to predict main and lateral bending angles. A summary of the above explanations is given in the revised manuscript.

- On the similar lines stated above, insufficient literature is presented to support the aim of the study. This point still needs further revision.

Appreciating the comments of the respected reviewer, the new references are added to the manuscript, and the following sentences are added in the Introduction section.

Support vector regression (SVR) is a learning algorithm implemented for the estimation of discrete values according to the principle of Support Vector Machines (SVM). SVM was introduced by Vladimir Vapnik [‎22] in 1992 and is a popular machine-learning tool for classification and regression. SVR is a powerful tool for estimation. Several pieces of research were published in different areas of mechanical engineering which utilized the SVR. SVR can predict drilling force in internal hole drilling of carbon fiber-reinforced polymer (CFRP) [‎23], wear prediction of cutting tools in the milling process [‎24], cutting force and temperature in bone drilling [‎25], cutting force and surface roughness in milling process [‎26] and turning process [‎27], prediction of laser cutting process cost of AISI316L stainless steel [‎28] and so on. By reviewing the different utilization of the SVR model and comparison of prediction results by other metaheuristic approaches (like ANN, GA, ANFIS, …) it can be concluded that the SVR model can predict the output parameters in complicated systems more accurately.

  1. Xu, C.; Yao, S.; Wang, G.; Wang, Y.; Xu, J. A prediction model of drilling force in CFRP internal chip removal hole drilling based on support vector regression. Int J Adv Manuf Technol 2021, 117, 1505–1516, doi.org/10.1007/s00170-021-07766-0.
  2. Benkedjouh, T.; Medjaher, K.; Zerhouni, N.; Rechak, S. Health assessment and life prediction of cutting tools based on support vector regression. J Intell Manuf 2015, 26, 213–223, doi: 10.1007/s10845-013-0774-6.
  3. Rabiee, A.H.; Tahmasbi, V.; Qasemi, M. Experimental evaluation, modeling and sensitivity analysis of temperature and cutting force in bone micro-milling using support vector regression and EFAST methods. Eng Appl Artif Intell 2023, 120, 105874, doi:1016/j.engappai.2023.105874.
  4. Yeganefar, A.; Niknam, S.A.; Asadi, R. The use of support vector machine, neural network, and regression analysis to predict and optimize surface roughness and cutting forces in milling. Int J Adv Manuf Technol 2019, 105, 951–965, doi: 10.1007/s00170-019-04227-7.
  5. Asilturk, I.; Kahramanli, H.; El Mounayri, H. Prediction of cutting forces and surface roughness using artificial neural network (ANN) and support vector regression (SVR) in turning 4140 steel. Mat Sci Technol 2012, 28, 980–986, doi: 10.1179/1743284712Y.0000000043.
  6. Jović, S.; Radović, A.; Šarkoćević, Ž.; Petković, D.; Alizamir, M. Estimation of the laser cutting operating cost by support vector regression methodology. Appl Phys A 2016, 122, 798, doi: 10.1007/s00339-016-0287-1.

- Results are presented in wrong way. Regression is used to find association between dependent and independent parameter. Figure 1 presents the actual measurements vs. predicted results by the SVR model for the main and lateral bending angles. Which one is dependent and independent parameter here? Actual and predicted values of same angle are plotted on X and Y axis. This is just a comparison and not regression.

Appreciating the insightful comments of the respected reviewer, and as rightly pointed out, regression is a statistical analysis method to identify the relationship between the variables. The relationship can be identified between the dependent and independent variables. In this study, the independent variables are the input variables of the model including laser power, laser beam diameter, scanning speed, irradiation length, irradiation scheme, and the number of irradiations. Also, the dependent variables are the output variables (main bending angle and lateral bending angle), which are estimated based on the inputs. The main goal of the presented work is to obtain a predictor model that can accurately estimate the desired outputs based on the input values. Many studies have been published focusing on obtaining the predictor model in different fields, so we decided to use the SVR model because of its advantages. In many similar studies that have used the SVR approach, it is found that to evaluate the model, they display the actual and predicted values in the form of a scatter plot. For example, we can refer to the cases listed in Table 1, all of which have benefited from support vector regression. However, it should be mentioned that the relationship between the input (independent) and output (dependent) variables can be found in the obtained model, which was not considered in this research. A summary of the above explanations is given in the revised manuscript.

Table 1. Examples of SVR model applications

No.

REFs

Comparison between

Scatter plot

1

REF [1]

Comparison between predicted and experimental sorption capacity of Lead

2

REF [2]

Comparison between predicted and measured tunnel boring machine penetration rates

3

REF [3]

Comparison between predicted and observed tool wear in face milling

4

REF [4]

Comparison between predicted and experimental cutting force in turning steel

5

REF [5]

Comparison between predicted and real building energy consumption

6

REF [6]

Comparison between predicted and actual operating cost

7

REF [7]

Comparison between predicted and actual adhesion Strength

[1]Parveen, N., S. Zaidi, and M. Danish, Support vector regression model for predicting the sorption capacity of lead (II). Perspectives in Science, 2016. 8: p. 629-631.

[2]Mahdevari, S., et al., A support vector regression model for predicting tunnel boring machine penetration rates. International Journal of Rock Mechanics and Mining Sciences, 2014. 72: p. 214-229.

[3]Bhattacharyya, P. and S. Sanadhya. Support vector regression based tool wear assessment in face milling. in 2006 IEEE International Conference on Industrial Technology. 2006. IEEE.

[4]Asilturk, I., H. Kahramanli, and H.E. Mounayri, Prediction of cutting forces and surface roughness using artificial neural network (ANN) and support vector regression (SVR) in turning 4140 steel. Materials Science and Technology, 2012. 28(8): p. 980-986.

[5]Zhong, H., et al., Vector field-based support vector regression for building energy consumption prediction. Applied Energy, 2019. 242: p. 403-414.

[6]Jović, S., et al., Estimation of the laser cutting operating cost by support vector regression methodology. Applied Physics A, 2016. 122: p. 1-5.

[7]Hazir, E., T. Ozcan, and K.H. Koç, Prediction of adhesion strength using extreme learning machine and support vector regression optimized with genetic algorithm. Arabian Journal for Science and Engineering, 2020. 45: p. 6985-7004.

- This paper doesn’t illustrate procedure of data collection, preparation, pre-processing, and processing in detail. Was the data normalized/ standardized?

 Thanks to the esteemed reviewer, there is a revised sub-section titled “2.1. Laser Tube Bending and data preparation” in the revised manuscript that shows the procedure of data collection and preparation. It should be noted that the data obtained by experimental tests are presented in Table A.1 Appendix of the revised manuscript. Also, the input data is normalized in the range of , which is explained in the revised manuscript.

- Additionally results must be provided considering different holdout % and holdout validation approach.

The following sentences were added to the manuscript to describe the holdout validation technique.

After implementing the experimental tests and measuring the output variable, the results should be split into two sub-sets for cross-validation. In this article, the holdout cross-validation technique (70% training data, 30% testing data) was used for evaluating the accuracy of the SVR model. So, the obtained results are randomly divided into two sub-sets.

- There is no discussion on results. The results are also included considering limited examples.

Appreciating the insightful comments of the respected reviewer, the main goal of the present work is to attain a predictor SVR model for estimating main and lateral bending angles in laser tube bending process that can be evaluated by different criteria. For this purpose, in the present manuscript, the results are presented in the form of figures 3-7, along with the relevant explanations.

- Was the algorithm trained using standard hyperparameters or were they altered?

As rightfully pointed out by the respected reviewer, in the presented predictor model, standard hyperparameters  (tuning parameter) and  (insensitivity loss function) are used. It is worth mentioning, hyperparameters   have been changed in the training phase, and in the final model, values (12, 0.1) and (9, 0,1) have been used to estimate main and lateral bending angles, respectively. The above description was added to the revised manuscript.

- Comment on computational time and complexity in training of algorithm.

According to the experimental nature of the present work and the number of available data (92 experimental tests), the computing time for training the network to predict the outputs are 0.2s, which was also added in the text of the revised manuscript.

- The manuscript is more like a report rather than a research paper failing in solid discussion. Revise results and discussion part by critically examining results and include inferences drawn. 

Appreciating the comments of the respected reviewer, based on the reviewers’ comments, we made many modifications to the manuscript. We hope that the revised manuscript will be approved by the respected reviewer as a research paper.

- How to ensure robustness of the model in high noisy environment?

Appreciating the scrutiny of the respected reviewer, the outputs in the present work (main and lateral bending angles) are completely calculated by mechanical method and using manual measuring tools. Therefore, due to not using the sensor and electrical signal, we did not consider the effect of environmental noise. However, if the respected reviewer deems it appropriate, the effect of noise can also be examined in the obtained results.

- How to deal with the data diversity of present moment and moment in the future?

Appreciating the comments of the respected reviewer, the model presented in this work is not a time series model, and the outputs are predicted based on the input values. At last, it is worth noting that the current work is the first attempt to implement the SVR method for the prediction of the main bending angle and lateral bending angle as a very complicated process variable that depends on six different input variables. The accuracy and precision of metaheuristic methods depend on the volume of investigation data. In this work, 92 sets of experiments have been carried out and the measured data were included in the analysis. Increasing the amount of data may lead to better prediction while the current model has acceptable accuracy. A summary of the above explanations is given in the revised manuscript.

Reviewer 2 Report

I have received the manuscript entitled "Developing a Support Vector Regression (SVR) Model for the prediction of main and lateral bending angles in the Laser Tube bending Process" for review and have found it to be well-written and quite interesting.

I made some minor linguistic, data presentation, and clarification corrections highlighted "in YELLOW colour" inside the pdf file. By double clicking on any highlighted text, the authors will find a correction or a suggestion or concern for clarification.

I am repeating some of these suggestions here (but in a more general manner):

1- I suggest adding a figure to section 2 in order to show the bending process schematic, showing a sample tube irradiated by a laser beam and also showing the process parameters and the output parameters.

2- It will be great if the authors explained the terms/symbols in the mathematical equations in a simple and an adequate manner.

3- The paper doesn’t include information about the software used to implement the models.. also the authors mentioned the implementation of a Box-Behnken design of experiments, but there are not details on the software used or the design matrix, etc…. the authors also mentioned data in (Table A.1 Appendix), but it was neither included in the manuscript nor in a supplementary document.

4- I highly suggest that the authors add parts of the experimental results in either a tabular form or as a figure. The readers should at least know what were the minimum and maximum values of the output parameters (bend angles), and how were these values measured… Also, a photo of a sample bent tube is good to give the readers a closer view on the results.

5- In the results section, I highly recommend that the authors (briefly) link their findings to physical behavior such as laser-tube interactions occurring during the bending process… the authors may briefly describe the heating regimes in the light of process parameters used (in a way similar to some sentences in the literature review section).

6- I suggest that the authors expand the discussion/conclusion section a little bit further, so that it focuses on the significance of the findings and their potential uses in the future.

7- In general and throughout the manuscript, I suggest trying to use shorter sentences, however, I believe it is sometimes difficult in studies like this one, but at least, the authors should try so that the reader have ease in following the results and arguments.

Apart from the above general remarks and the detailed comments in the highlighted text within the pdf.... I enjoyed reading the work and believe it merits a valuable publication. Most of my suggestions are aimed to make the manuscript more readable and to magnify the impact of the results.

Author Response

 The authors greatly appreciate the reviewer for their insightful comments and suggestions. According to the reviewer comments, the manuscript has been edited. The comments were sorted, and the relevant answers were highlighted in green color as follows. The necessary changes in the manuscript have been shown in red color.

I have received the manuscript entitled "Developing a Support Vector Regression (SVR) Model for the prediction of main and lateral bending angles in the Laser Tube bending Process" for review and have found it to be well-written and quite interesting.

- I made some minor linguistic, data presentation, and clarification corrections highlighted "in YELLOW colour" inside the pdf file. By double clicking on any highlighted text, the authors will find a correction or a suggestion or concern for clarification.

 Many thanks to the respected reviewer for careful reading and leading comments in the PDF file. All of the hints are applied in the manuscript.

I am repeating some of these suggestions here (but in a more general manner):

1- I suggest adding a figure to section 2 in order to show the bending process schematic, showing a sample tube irradiated by a laser beam and also showing the process parameters and the output parameters.

The following sentence and Fig. 1 are added to the manuscript.

Fig.1 shows a schematic view of LTBP with the definition of the main bending angle along the vertical direction and lateral bending angle in the horizontal plane.

Figure 1. Schematic view of LTBP and definition of main and lateral bending angles.

2- It will be great if the authors explained the terms/symbols in the mathematical equations in a simple and an adequate manner.

The section describing the basics of SVR is revised and after each parameter, a description is added also.

3- The paper doesn’t include information about the software used to implement the models. also the authors mentioned the implementation of a Box-Behnken design of experiments, but there are not details on the software used or the design matrix, etc…. the authors also mentioned data in (Table A.1 Appendix), but it was neither included in the manuscript nor in a supplementary document.

The RSM has been implemented by Minitab software and the SVR models are developed in Python software. The following sentences are added accordingly. Also, Table A.1 Appendix is added to the manuscript after the Reference section.

The SVR models are developed with Python Software.

Minitab software was used for the design of experiments and data analysis in published work [‎28].

4- I highly suggest that the authors add parts of the experimental results in either a tabular form or as a figure. The readers should at least know what were the minimum and maximum values of the output parameters (bend angles), and how were these values measured… Also, a photo of a sample bent tube is good to give the readers a closer view on the results.

Fig. 1, Fig. 2, and Table A.1 are added to the manuscript and the readers can see the range of output parameters. The details about the experimental measurement can be found in Ref. [29] and due to similarity check limits which should be less than 20 %, more details cannot be described here. The following sentences guide the readers for more details.

It is recommended the readers read the previous publication of the authors [‎23] for additional details and then read this article.

More details of the experimental setup and definitions are presented in Ref. [‎23] and are not repeated in this article.

5- In the results section, I highly recommend that the authors (briefly) link their findings to physical behavior such as laser-tube interactions occurring during the bending process… the authors may briefly describe the heating regimes in the light of process parameters used (in a way similar to some sentences in the literature review section).

The SVR model can only predict the output parameters by a sophisticated link between inputs and outputs. So, interpreting the physical aspect of output variation is not possible now. The interactions of process parameters are discussed in Ref. [29]. The authors keep in mind the proposed idea and try to find a way to link these two objects and publish it as soon as possible.

6- I suggest that the authors expand the discussion/conclusion section a little bit further, so that it focuses on the significance of the findings and their potential uses in the future.

Several sentences were added to the results and discussion section and also in the Introduction section for describing the aims and significance of the SVR model which are highlighted in red color.

7- In general and throughout the manuscript, I suggest trying to use shorter sentences, however, I believe it is sometimes difficult in studies like this one, but at least, the authors should try so that the reader have ease in following the results and arguments.

 The authors tried to edit some long sentences but in some cases, it was not possible. One of the limitations is increasing the similarity of the manuscript with previously published works. But the authors tried and some sentences are edited.

A comment in PDF file: looking at MAE and RMSE values in table2, I can see that lateral bending results are more accurate.

Appreciating the comments of the respected reviewer, as can be seen in Table A.1 Appendix, the main bending angle values are larger compared to the lateral bending angle, so they have bigger MAE and RMSE errors. It is better to refer to the MAPE and R2 criteria to compare the approximation accuracy, which shows the normalized error and determination factor. According to the values presented in Table 2, it can be seen that the SVR model performs better in predicting the main bending angle compared to the lateral bending angle due to the lower MAPE value and higher R2.

Apart from the above general remarks and the detailed comments in the highlighted text within the pdf.... I enjoyed reading the work and believe it merits a valuable publication. Most of my suggestions are aimed to make the manuscript more readable and to magnify the impact of the results.

The authors gratefully thank the respected reviewer for invaluable comments about the approach and details. It is tried to apply all of the comments and hopes it will be done.

Reviewer 3 Report

This paper is a good study about the viability of the support vector regression model to predict the laser conditions to bend the pipes. Several predict tools were adequate employed to carry out this modelling. The model was correctly validated with a good comparison between experimental and predicted data. This work can be so use for the industry. Thus, this paper should be published in the journal. However, the authors should consider the specific comments to improve this paper.

Comments;

To add reference in line 28 "section and ovalization as an unwanted phenomenon appears [REF]. The laser beam has severa"

To include reference in line 30 "irradiated heating can be controlled very easily [REF]. The cost of die fabrication will be waived "

To incorporate reference in line 39 "with an analytical solution or statistical prediction [REF]."

To move Artificial Neural Network (line 50) to line 46 "eters of the forming process. Then the trained [HERE] ANN helped to increase the correlation"

To add reference in line 61 "analytical model [REF]. Fetene et al. [8] studied the effects of the width and thickness of the sheet"

The equation should be referred if they were not developed by the researchers to avoid plagiarism demands. 

To define the acronym MAPE in line 231 "Fig. 2 and 3 illustrate the absolute error and MAPE error for the main bending angle"

Author Response

The authors greatly appreciate the reviewer for their insightful comments and suggestions. According to the reviewer comments, the manuscript has been edited. The comments were sorted, and the relevant answers were highlighted in green color as follows. The necessary changes in the manuscript have been shown in red color.

This paper is a good study about the viability of the support vector regression model to predict the laser conditions to bend the pipes. Several predict tools were adequate employed to carry out this modelling. The model was correctly validated with a good comparison between experimental and predicted data. This work can be so use for the industry. Thus, this paper should be published in the journal. However, the authors should consider the specific comments to improve this paper.

Comments;

- To add reference in line 28 "section and ovalization as an unwanted phenomenon appears [REF]. The laser beam has severa"

New reference is added to the manuscript as follows.

“… ovalization as an unwanted phenomenon appears [‎1]. The …”

  1. Safari, M.; Alves de Sousa, R.; Joudaki, J. Experimental investigation of the effects of irradiating schemes in laser tube bending process. Metals 2021, 11, 1123, doi: 3390/met11071123.

To include reference in line 30 "irradiated heating can be controlled very easily [REF]. The cost of die fabrication will be waived "

New reference is added to the manuscript as follows.

“ … the irradiated heat/energy can be controlled very easily [‎2]. The cost … “

  1. Safari, M.; Alves de Sousa, R.; Joudaki, J. Recent Advances in the Laser Forming Process: A Review. Metals 2020, 10, 1472, doi: 3390/met10111472.

To incorporate reference in line 39 "with an analytical solution or statistical prediction [REF]."

New reference is added to the manuscript as follows.

“… with an analytical solution or statistical prediction [3]. “

  1. Ponticelli, G.S.; Guarino, S.; Giannini, O.; Tagliaferri, F.; Venettacci, S.; Ucciardello, N.; Baiocco, G. Springback control in laser-assisted bending manufacturing process by using a fuzzy uncertain model. CIRP 2020, 88, 491–496, doi: 10.1016/j.procir.2020.05.085.

To move Artificial Neural Network (line 50) to line 46 "eters of the forming process. Then the trained [HERE] ANN helped to increase the correlation"

The acronym of ANN was defined in line 44 accordingly. The comment is applied in the manuscript.

To add reference in line 61 "analytical model [REF]. Fetene et al. [8] studied the effects of the width and thickness of the sheet"

The sentence was part of the following paragraph which describes the published work of Imhan et al. [10]. So, there is no need for adding additional references.

Imhan et al. [‎10] investigated the laser bending of stainless steel 304 tubes. The bending angle was calculated by an analytical model. The analytical model considered the variation of material properties by temperature rise. Then, Particle Swarm Optimization (PSO) was utilized for the optimization of the results and to decrease the mean absolute error. The PSO optimization helped to determine values for the material specification (density, yield stress, elastic modulus, specific heat, coefficient of thermal expansion(CTE)) such that the effect of temperature increase was included in the analytical model.

The equation should be referred if they were not developed by the researchers to avoid plagiarism demands.

The following sentence is added to the manuscript.

The equations of the SVR model were derived by Vapnik [‎22], Patel et al. [‎30], and Huang and Tsai [‎31].

To define the acronym MAPE in line 231 "Fig. 2 and 3 illustrate the absolute error and MAPE error for the main bending angle"

The captions of Fig. 2 and 3 are edited by defining the acronym of MAPE.

Figure 4. a) absolute error, b) mean absolute percentage error (MAPE) for all results corresponding with the main bending angle.

Figure 5. a) absolute error, b) mean absolute percentage error (MAPE) for all results corresponding with lateral bending angle.

Round 2

Reviewer 1 Report

The authors have addressed all my comments positively.